# Functionality and Health-Promoting Properties of Polysaccharide and Plant-Derived Substances from *Mesona chinensis*

**DOI:** 10.3390/foods13071134

**Published:** 2024-04-08

**Authors:** Romson Seah, Sunisa Siripongvutikorn, Santad Wichienchot, Worapong Usawakesmanee

**Affiliations:** 1Department of Chemistry, Faculty of Education, Fatoni University, Yarang, Pattani 94160, Thailand; romson.seah@ftu.ac.th; 2Centre of Excellence in Functional Foods and Gastronomy, Faculty of Agro-Industry Prince of Songkla University, Hat Yai, Songkhla 90110, Thailand; santad.w@psu.ac.th (S.W.); worapong.u@psu.ac.th (W.U.)

**Keywords:** *Mesona chinensis*, medicinal plant, pharmacological activity, bioactive compound, polysaccharide

## Abstract

*Mesona chinensis*, in Thai called Chao Kuay and in Chinese Hsian-tsao, belongs to the Lamiaceae family. This herbal plant grows widely in Southern China, Taiwan (China), Malaysia, the Philippines, Indonesia, Vietnam, and Thailand. The Mesona plant is used to make functional products such as drinks and soft textured sweet treats, and also traditional medicine, to treat heat stroke, high blood pressure, heart attack, high blood sugar, hepatic diseases, colon diseases, inflammatory conditions, and to alleviate myalgia. The proximate composition of *M. chinensis* is a mixture of protein, fat, fiber, ash, and minerals. The main biological compounds in *M. chinensis* extracts are polysaccharides, terpenoids, flavonoids, and polyphenols, with wide-ranging pharmacological properties including antioxidant, antidiabetic, antilipidemic, carcinoma-inhibitory, renal-protective, antihypertensive, DNA damage-protective, and anti-inflammatory effects. This review investigated the proximate composition, polysaccharide type, and pharmacological properties of *M. chinensis* extracts. Phytochemical properties enhance the actions of the gut microbiota and improve health benefits. This review assessed the functional and medicinal activities of *M. chinensis* extracts. Future studies should further elucidate the in vitro/in vivo mechanisms of this plant extract and its impact on gut health.

## 1. Introduction

The herbaceous plant *Mesona chinensis* is part of the Lamiaceae family [1]. It grows well in southern China (in Zhejiang, Jiangxi, and Guangdong provinces, as well as Taiwan) [2], India, Malaysia, the Philippines, Indonesia, Vietnam, and Thailand [3,4]. The Mesona plant has several names, including *Mesona chinensis* Benth., *Mesona procumbens* Hemsl., *Mesona parviflora* (Benth.) Briq., *Mesona philippinensis* Merr., *Mesona palustris* Blume., *Mesona wallichiana* Benth., *Mesona elegans* Hayata., *Geniosporum parviflorum* Benth., *Platostoma chinense* (Benth.) A. J. Paton., and *Platostoma palustre* (Blume) A. J. Paton [5,6,7]. *Mesona chinensis*, *Mesona procumbens*, and *Mesona palustris* have been documented in China, Taiwan (China), and Indonesia, respectively. The plant has local names in different countries. For example, in China and Taiwan (China), it is Hsian-tsao or Liangfen Cao; in Indonesia and Malaysia, it is black cincau; in Vietnam, it is Suong Sao; and in Thailand it is Chao Kuay [8,9,10,11]. This perennial herb grows 15–100 cm high and its stem is covered with soft hair. The leaves are narrowly ovate and almost circular [12], as demonstrated in Figure 1. The plant grows in ditches, forest slopes, streams, and also on dry sandy land, demonstrating good environmental adaptability [13]. However, drought in summer and cold winters lead to poor plant growth or even death, and impact the yield [14], with a reduced gross weight, number of roots, and aerial parts [15]. This herb is normally digested as a functional beverage and a semi-solid sweet dessert and has been used as a traditional medicine in China, Vietnam, and Indonesia for thousands of years [2,16,17]. The plant is used to treat heat shock, fever, hypertension, heart attack, diabetes, muscle pains, liver diseases, colon diseases, and inflammatory conditions, and also to alleviate muscle or joint pain [16,18,19,20,21,22], because of the various active phytochemicals it contains, such as flavonoids and phenolic acid [23]. Mesona is also utilized to make a popular jelly dessert, due to its high polysaccharide component [24].

Many researchers have demonstrated the biological activities of phytochemical compounds. However, the nutraceutical and pharmacological properties reported for *M. chinensis* as evidence for its health benefits as a medicine are still limited. This review discussed the recent scientific information on the Mesona plant regarding the composition of its phytochemicals and its alteration of gut microbiota, as well as its health benefits. Several bioactive compounds and polysaccharide components have been reported to have various pharmacological effects including antioxidant activities [25], hypoglycemic and hypolipidemic activities [26,27], antiproliferative activities [28], a growth inhibitory effect on hepatocellular carcinoma (HepG2) cells [15], anti-dyslipidemia activities [29], renal-protective activities [30], antihypertensive [31] and DNA damage-protective activities [32], anti-inflammatory activities [33], antimutagenic effects [34], and antibiosis activities [35].

## 2. Proximate and Mineral Composition

The proximate and mineral components of *M. chinensis* leaves are presented in Table 1. Differences in the environment [36] and climatic conditions impact major plant constituents and chemical components [37]. The mineral contents of *M. chinensis* when extracted with sodium bicarbonate in heated water were 1420 ± 10 µg/g Mg, 5.8 ± 1 µg/g Cu, 26 ± 1 µg/g Zn, 66 ± 1 µg/g Fe, 290 ± 5 µg/g Mn, 2810 ± 10 µg/g Ca, 10,600 ± 20 µg/g K, and 40,300 ± 10 µg/g Na [36]. The major minerals in *M. chinensis* gum (MBG) included Na, K, Ca, and Mg, with Na making up 73% of the total. Similarly, Yuris et al. stated that polysaccharide extracted from plant powder consisted of K, Mg, P, Ca, Na, Fe, Cu, I, Mn, Zn, and Se [38].

## 3. Polysaccharides and Phytochemicals of *M. chinensis* Extracts

Polymeric macromolecules of carbohydrates or polysaccharides contain long chains of monosaccharide units joined by glycosidic linkages [42,43]. Polysaccharides are essential for all living cells, particularly in plants and microorganisms such as bacteria, yeast, and mold [44,45]. Polysaccharides can be divided into two forms, including homopolysaccharides, which comprise only one type of simple carbohydrate, and heteroglycans, which contain a mixture of two or more different monosaccharides [46].

### 3.1. Physicochemical Properties

Polysaccharides comprise the main functional components of *M. chinensis* [47], which include structural heteropolysaccharide consisting of an α-1, 4-linked galacturonan backbone with some α-1, 2-linked rha*p* (Rhamnose) residues (Figure 2), and large amounts of uronic acid [20,48], similar to the pectin structure. Wang et al. reported that polysaccharides of the Mesona plant are acidic glycoprotein compounds containing uronic acid and some protein [49]. Mesona plants can be classified as acidic heteroglycan material containing galacturonic acids and a few glucuronic acids in the polysaccharide backbone [50]. The monosaccharide composition comprised glucose, galactose, galacturonic acid, rhamnose, arabinose, and xylose, while mannose was also reported, as shown in Table 2. Chen et al. listed the different monosaccharide compositions of seven freshly dried and six dried Mesona plants after 1 year of storage [51]. Their results indicated that storage time was an important factor affecting monosaccharide quality, while monosaccharide compositions were dependent on the material source, extraction method, growth stage of the plant, and the cultivation environment [25,52,53]. Previous FT-IR spectral band and peak studies determined that the structure of Mesona polysaccharides (MPs) consisted of the following three functional groups: carbonyl (C=O), carboxyl (COOH), and hydroxyl (OH) [49,50,54,55]. *M. chinensis* has been widely studied for its chemical composition and various biological activities in China and Taiwan (China), but few studies are documented from the Indonesian region, possibly due to the lack of scientific records, research funding support, and marketing value. Black jelly, as a product, has been mainly produced from Mesona plants cultivated in China; however, recently, some large companies producing black jelly (or Chao Kuay in Thailand) have used raw materials from several regions, including Vietnam and Indonesia, due to the better pricing and more consistent quality between each lot (according to survey and interview data from company owners in Thailand and Malaysia). 

### 3.2. Phytochemicals

The powder extraction of *M. chinensis*, using methanol to separate it into acidic ethyl acetate fractions, was evaluated for polyphenolics. The results showed that the polyphenolic compounds found included protocatechuic acid, *p*-hydroxybenzoic acid, vanillic acid, caffeic acid, and syringic acid [18]. The ethanolic *M. chinensis* extraction yielded rosmarinic acid, apigenin, 7-hydroxycoumarin, ferulic acid, and rutin [16]. Eleven novel diterpenoids were reported as resulting from the methanolic extraction of *M. chinensis* after isolation with *n*-hexane and CH_2_Cl_2_, including seven *ent*-kauranes, three *ent*-atisanes, and one sarcopetalane [28]. Bioactive compounds of *M. chinensis* ethanolic extracts and their fractions, named as F0, F10, F20, F30, F40, F50, and MTFs (*Mesona* total flavonoids) (including aqueous extract (AE)) and detected using high performance liquid chromatography with mass spectrometry (HPLC-MS) analysis, mainly contained caffeic acid, quercetin 3-O-galactoside, isoquercetin, astragalin, rosmarinic acid, aromadendrin-3-O-rutinoside, rosmarinic acid-3-O-glucoside, and kaempferol-7-O-glucoside. MTFs, prepared using an ethanolic extract and X-5 macroporous resin as purification for flavonoid enrichment, exhibited the highest peaks of these compounds [27]. *M. chinensis* extracted in boiled water provided crude polysaccharides (301.7 mg/g) and β-1,3-glucang (68.9 mg/g). The functional groups of the crude polysaccharide extract were confirmed, using FT-IR spectrophotometry, to be hydroxyl and carbonyl [11]. Another study of Mesona plants, gathered in Southern China and extracted using deionized water at 80 °C for 2 h before ultra-high performance liquid chromatography with quadrupole time-of-flight-mass spectrometry (UPLC-Q-TOF-MS/MS) analysis, found 5757 compounds including 45 polyphenols, 6 terpenoids, and 6 other unknown compounds [56]. Water extracts of *M. chinensis* recorded seven phenolic compounds, namely kaempferol, apigenin, caffeic acid, protocatechuic acid, syringic acid, vanillic acid, and *p*-hydroxybenzoic acid. Interestingly, caffeic acid and kaempferol gave the highest values of phenolic constituents in the extracts [57]. The results indicated that extraction media with different polarities identified bioactive compounds by following the “like dissolves like” rule. However, raw material types, planting area and harvesting time, preparation method, extraction condition, and different determination assays can also give diverse results.

## 4. Pharmacological Properties

### 4.1. Pharmacological Properties

Consumers now prefer to eat natural plant food to promote their nutrition, as this contains pharmaceuticals, fibers, pigments, minerals, vitamins, and unsaturated fatty acids and is free of manufactured food additives [58]. *M. chinensis* polysaccharides (MCPs) are attracting interest, due to their potential biological activities in food and their pharmacological properties [59], as shown in Figure 3. Mesona has been applied into extruded rice products [60] and effervescent powder [29], encapsulated in alginate beads [61], and used for its antitumor [62], anticoagulant [63], antioxidant [64], antidiabetic [65,66], and immunomodulatory activities [67,68,69]. Most extracted polysaccharides from medicinal plants are nonpoisonous and have no adverse issues [70,71]. Natural products are preferred to synthetic agents, which typically exhibit negative side effects.

### 4.2. Antioxidant Activity

Free radicals, high-energy particles that ricochet widely and damage cells, can be created in living cells as highly unstable molecules, such as reactive oxygen species (ROS), comprising superoxides, hydroxyls, peroxyls, and alkoxyls [72]. ROS induce various chronic and degenerative diseases, including dementia and shaking palsy [73], respiratory, neurodegenerative, and digestive diseases [74], cancer, diabetes mellitus, insulin resistance, cardiovascular diseases, atherosclerosis, and aging [75]. MCPs exhibited high potential antioxidant activities. Lai et al. reported that polysaccharide gum from Mesona leaf was strongly concentration-dependent on free radical scavenging activities [40]. The 2,2-diphenyl-1-picrylhydrazyl (DPPH) free radical scavenging IC_50_ value was 68.6% at 1250 µg/mL. Superoxide free radical scavenging activities increased with the extract concentration (86.5% at a dose level of 1250 µg/mL), while the chelating ability of ferrous ion also increased to 74.4% at 1.5 mg/mL, and the reducing power increased in a dose-dependent manner. FeSO_4_-H_2_O_2_ was used to induce malondialdehyde (MDA) in rat histology and was evaluated for lipid peroxidation. MDA formation in rat tissue homogenate (brain, liver, and heart tissue) significantly decreased by 15.86–83.68% when adding 5–40 mg/mL of polysaccharide concentrate. Interestingly, water-soluble polysaccharides showed higher potential in hepatic and heart organs than the brain [40], while MCPs provided strong hydroxyl radicals in a concentration-dependent manner, with the highest scavenging rate being 54.36 ± 1.56% at 1600 µg/mL. By contrast, the superoxide anion scavenging activity reached 58.42 ± 1.17%, when the value of MCPs was 1600 µg/mL. The scavenging activities of MCPs on DPPH free radicals gradually increased (55.59 ± 0.69%) as MCP concentration increased to 1600 µg/mL [50].

Mesona polysaccharides (MPs) at 1000 μg/mL showed high DPPH free radical scavenging activity of 75.11 ± 0.31%. Similarly, MPs also showed scavenging effects (63.26 ± 0.28%) against hydroxyl radicals and demonstrated significant dose-dependent defense against H_2_O_2_-promoted injury to RAW 264.7 macrophage cell line at 100 μg/mL, measured as 78.58 ± 0.11%. Oxidative damage caused by lipid peroxidation to RAW 264.7 cells, measured using the MDA assay gradually decreased to 96.88 ± 2.52 μmol/mL at a 100 μg/mL MP concentration [55]. Chen et al. studied the ability of MPs to scavenge free radicals using the DPPH and ABTS (2,2′-azino-bis (3-ethylbenzthiazoline-6-sulphonic acid)) assays. Their results demonstrated DPPH antioxidant activities of 75.59 ± 0.13% at 1000 μg/mL, while the ABTS free radical scavenging IC_50_ value was 332.34 μg/mL [51].

The antioxidant activities of aqueous extracts of *M. chinensis* from various areas in Southern China, including Guangdong Meizhou, Guangdong Raoping, Guangdong Shaoguan, Fujian Longyan, Fujian Zhangzhou, Jiangxi Jian, Jiangxi Ganzhou, Guangxi Chongzuo, and Guangxi Yulin, were evaluated. The results showed that samples obtained from Guangdong Raoping provided the strongest antioxidant ability (as measured using DPPH and ABTS assays) with IC_50_ values of 0.00076 ± 0.00006 mg/mL and 0.00383 ± 0.00017 mg/mL, respectively [56], indicating that planting area or physiological race significantly impacted biological activity, possibly due to the quality and quality of the phytochemical compounds. Recently, MCP aqueous and ethanolic extracts and their fractions (F0, F10, F20, F30, F40, and F50) and MTFs were intensively evaluated using DPPH free radical scavenging and FRAP (Ferric reducing antioxidant power) assays. MTFs and F30 exhibited higher free radical scavenging abilities and reducing power than the control (vitamin C), with IC_50_ values of 0.005323 mg/mL and 0.005278 mg/mL, respectively [27].

Polyphenols and antioxidant activities were further evaluated through gastrointestinal digestion experiments. An aqueous dried *M. chinensis* extract was encapsulated in alginate beads to study the polyphenol and antioxidant activities under a gastrointestinal digestion experiment. The release of total phenolic content (TPC) from the beads was low, at only 8.9%, after soaking in water for 4 h. TPC and FRAP activities of the encapsulated beads were higher than the control, because the gastric pH was lower than the pKa value of the bead material as an alginate monomer; therefore, the protective effect was strong [61]. The results indicated that the Mesona extract improved health through antioxidation activity, particularly when encapsulation was applied before digestion.

### 4.3. Cancer and Toxicity Studies

Cancer is the most common cause of death in Thailand. Natural products have attracted an increasing interest as novel anticancer drugs, with potential biological activities associated with therapies that have fewer lower side effects [76,77]. The cytotoxic effects of different extractions (water, ethanol, and ethyl acetate) from *M. chinensis* were tested for anticancer activity against Hela cells in an in vitro culture. The results indicated that the water extract induced higher anticancer activity than the other two organic extracts (ethanol and ethyl acetate), with cytotoxicity against Hela cells giving IC_50_ values of 0.1326, 0.146, and 0.18296 mg/mL, respectively [78]. In ana in vivo study, MCPs acted in a dose-dependent manner (0.10, 0.20, and 0.30 g/kg body weight) and provided hepatoprotective activity against acute liver damage induced by CCl_4_. A significant reduction into serum markers was found in the livers of mice after treatment with MCPs at medium and high doses (0.20 and 0.30 mg/kg body weight), when assayed through aspartate aminotransferase (AST) and alanine aminotransferase (ALT) parameters. A reduced impact of CCl_4_ toxicity on the serum markers (aspartate aminotransferase, AST, and alanine aminotransferase, ALT) of liver damage in mice was noticed at medium and high doses of MCPs. MCPs also increased levels of antioxidant enzymes (superoxide dismutase, SOD) and non-enzyme antioxidants (glutathione, GSH), while lipid peroxidation levels of liver tissues, evaluated by MDA, significantly declined. Serum levels of IL-1*β* and TNF-*α* increased, indicating that MCPs had hepatoprotective activity against acute injured liver induced by CCl_4_ [49].

Ethanolic extracts of *M. chinensis*, at concentrations of 1560–100,000 µg/mL and 390–3130 µg/mL, showed significant inhibition effects against the viability of CT-26 (colorectal cancer cell line) and HT-29 (Human colon cancer cells) colorectal cancer cell lines, respectively [79]. The protective effect of a methanolic extract of *M. chinensis* on human leukemia cancer cells was reported by Huang et al. [28]. The methanolic extract revealed 11 new diterpenoid compounds, mesonols 1–11, and mesonols 1–4 compounds, which provided antiproliferative activities against five cancer cell lines (human lung carcinoma (A549), human liver carcinoma (Hep-3B), human prostate carcinoma (PC-3), human colon carcinoma (HT29), and human monoblastic leukemia (U937)), with reduced toxicity against the RAW 264.7 cell line. Mesonols 1–2 showed higher antiproliferative activities against U937 cells than the standard drug Camptothecin (CPT-11; irinotecan), with IC_50_ values of 2.66, 1.97, and 4.95 μM, respectively. The results suggested that Mesona plants could be used as alternative active ingredients and food for medicinal and nutraceutical purposes. However, further clinical trials are needed.

### 4.4. Hypolipidemic Effect

Hyperlipidemia or dyslipidemia is the primary cause of heart attack, resulting from increased serum levels of TC (total cholesterol), LDL (low-density lipoprotein), or TG (triglycerides), or from decreased amounts of HDL (high-density lipoprotein) [80,81]. Pharmaceutical herbs are commonly utilized in various congestive heart failure treatments [82]. Handayani et al. stated that a semisolid product from *M chinensis* exhibited antihyperlipidemic activity in rats fed a high-density lipoprotein diet. Lipid profiles, including plasma cholesterol and triacylglycerol, were assessed using the enzymatic cholesterol oxidase-*p*-aminophenazone (CHOD-PAP) assay in Wistar rats induced with a hypercholesterol diet. *M. chinensis*, as an effervescent powder, was introduced to the treated rats for 3 weeks. The results showed that increasing doses of *M. chinensis* as an effervescent powder significantly influenced plasma lipid reduction [29]. The antihyperlipidemic activity of a Mesona ethanolic extract was tested on overweight mice fed with greasy food. The mice were separated into two groups. The first group (normal weight mice) was fed with an ethanolic extract of *M. chinensis* (0.40 g/kg body weight) and high-fat feed for 4 weeks, while the second (obese mice) was fed with an ethanolic extract of *M. chinensis* (0.40 g/kg body weight) and then reared on a high-fat feed for 15 days. Mice receiving NaCl 0.9% were used as the positive control, with mice receiving fenofibrate comprising the negative control group. After 4 weeks of plant extract treatment, the prevention mice group showed significantly lower TG concentration and total cholesterol/high density lipoprotein cholesterol (TC/HDL-C) levels than the control group. However, the mice receiving the plant extract did not show any significant differences in blood fat composition, compared with mice receiving fenofibrate and NaCl. The ethanolic extraction from *M. chinensis* gave protection to high-cholesterol mice [83].

Isolated flavonoid compounds, from an *M. chinensis* ethanolic extract at a concentration of 200 μg/mL, decreased the fat increment in oleic acid (OA)-induced HepG2 cells and inhibited compound C on 5′ adenosine monophosphate-activated protein kinase (AMPK). The glucose utilization of insulin induced HepG2 cells was significantly increased by MTFs and F30, compared to Metformin, which was used as the positive control [27]. The results indicated that the Mesona extract provided overweight control in a similar way to the mechanism of Metformin, which is currently used as the standard drug to treat obesity.

### 4.5. Hypoglycemic Effect

Diabetes mellitus (DM) is a chronic disease that causes es excessive blood sugar levels [84] caused by insulin secretion, insulin dysfunction, and/or both [85]. DM can be classified as insulin-dependent diabetes mellitus (IDDM) or type 1, which caused by autoimmune β-cell damage in the pancreas that leads to absolute insulin deficiency [86], or as non-insulin-dependent diabetes mellitus (NIDDM) or type 2, which is a metabolic disorder with variable phenotypic expressions, including β-cell insufficiency and insulin resistance [87]. The hypoglycemic effects and antioxidant activities generated by a diet of excessive calories in obese subjects were examined. The results indicated that *M. chinensis* (MC) extract, produced by boiling in distilled water, suppressed intestinal maltase and sucrase, with IC_50_ values of 4660 ± 220 µg/mL and 1300 ± 430 µg/mL, respectively. In contrast, the plant extract did not show inhibitory activity against pancreatic α-amylase. Interestingly, 1000 mg of MC extract with a HC (high carbohydrate) diet reduced postprandial plasma glucose, triglyceride, and MDA levels, while an increase in the plasma antioxidant capacity (FRAP and oxygen radical absorbance capacity, ORAC) of overweight subjects was noticed after treatment with MC [88]. Lim et al. examined the effect of MC on healthy Chinese men. They found that both gel and solution forms significantly reduced glycemic and insulinemic properties, compared to the control group (who received only glucose without MC extract) [23]. 

An extraction of MC collected from Guangdong Raoping in Southern China exhibited the highest inhibitory effect on the *α*-glucosidase enzyme, with an IC_50_ value of 35.05 ± 2.16 μg/mL [56]. The inhibition of the *α*-glucosidase enzyme enhanced carbohydrate digestion abilities, due to decreased blood glucose levels. Compounds with lower IC_50_ values resulted in higher anticarbohydrate digestibility. The results indicated that the topography of each area impacted the biological activities of the phytochemicals and bioactive compounds.

### 4.6. Renal Protective Activity

Hsian-tsao or *M. chinensis* water extract was given to ten Sprague Dawley female rats, together with an injection of streptozotocin (STZ) to induce diabetes. The expression of thrombospondin-1 (TSP-1) in the kidney was measured using immunohistochemistry, with significantly lower results in the plant-treated group than in the diabetic group. Kidney ultrastructural changes, assayed by transmission electron microscopy, were significantly less severe in the plant-treated group compared with the diabetic group, indicating that MC protected the kidneys of diabetic rats [21].

### 4.7. Inflammatory Activity

Inflammation contributes to several illnesses, including rheumatoid arthritis, atherosclerosis, and asthma, by stimulating the immune system, circulation system, and various organelles within the injured cells [89]. The inflammatory mechanisms induce the up-regulation of a series of pro-inflammatory cytokines, including interleukin IL-1, tumor necrosis factor (TNF), interferon (INF)-c, IL-6, IL-12, IL-18, and the granulocyte-macrophage colony-activating factor [90]. The inhibitory effects of ethanolic and water extracts of *M. chinensis* on methylglyoxal (MG)-induced glycation in mice were investigated, using Western blot to analyze IL-8, MIP-2, and MCP-1inflammation-related factors of RAW 264.7 cells. The MG-induced protein expression of IL-8, MIP-2, and MCP-1 in RAW 264.7 cells were notably reduced with ethanolic and water *M. chinensis* extracts (0.0125–0.025 mg/mL), compared with the control. Interestingly, the water extract exhibited higher anti-inflammatory activity against MG-induced inflammation [11]. The biological actions of specific MCP water extracts may play more important roles than phenolic compounds of ethanolic extracts, due to the structure and chemical composition of polysaccharides, which have at least two functional groups, such as C=O, -OH, COOH, -NR_2_, and -SH [91,92]. MCPs also had high uronic acid and protein contents, which may increase antioxidant activity and inhibit liver inflammation [49,93]. However, animal and clinical trials must be further performed, to gain strong evidence and scientific support.

MC polysaccharides (MCPs), extracted with boiled water at 100 °C for 2 h, alleviated pathological signs of DSS (dextran sulfate sodium)-induced colitis in mice, based on a reduction in body weight, an increase in colon length, and reduced disease activity index (DAI) scores. MCPs also improved inflammatory cell infiltration, disorganized glandular arrangement, and disrupted the intestinal structure in colonic tissues. Inflammatory cytokines showed significant increases in TNF-*α* and IL-1*β* levels and a decrease in IL-10 content. After DSS supplementation in the model control group, MCP administration reversed these expressions. MCP doses of over 0.2 g/kg/day significantly reduced the expression of phosphorylated proteins caused by MAPK/NF-*κ*B signaling pathways, as analyzed by Western blot analysis [94].

### 4.8. Gut Microbiota

The human body is lacking in gastrointestinal enzymes, and most polysaccharides cannot be directly digested by the stomach and small intestine [95,96]. Compounds that reach the large intestine are fermented by intestinal microbiota to generate short-chain fatty acids (SCFAs) and other metabolites, which play a key role in well-being [97,98,99]. MCPs administrated to mice at doses of 0.30 g/kg/day decreased the abundance of *Firmicutes*, *Verrucomicrobiota*, and *Proteobacteria* microbiota, but the incidence of *Bacteroidetes* increased. MCPs at high-dose levels increased *Bifidobacterium*, *Lactobacillus*, *Coprococcus*, and *Oscillospira*, while *Akkermansia*, *Clostridiaceae*, *Clostridium*, *Helicobacter*, and *Prevotella* were decreased. DSS-induced colitis in mice was improved compared with the NC (normal control) group without DSS-induced colitis. The dominant bacterial composition of the intestinal microbiota was identified using the linear discriminant analysis effect size (LEfSe) of *M. chinensis* Benth polysaccharides (MBP) treatment in a dose-dependent manner as follows: *Coprococcus* (with low dose, LD), *Akkermansia* and *Sphingomonas* (with medium dose, MD), and *Blautia* and *Dysgonomonas* (with high dose, HD) [94]. After injecting mice with CTX (cyclophosphamide) to induce liver damage and treating them with MCP concentrations ranging from 0.05 to 0.20 g/kg bw for 7 days, the results showed that, at high dosage levels (0.2 g/kg bw), serum aspartate aminotransferase (AST) and alanine aminotransferase (ALT) were controlled, while antioxidant activity, repaired liver damage, the inhibition of inflammatory cytokines in the liver, and the concentrations of lipopolysaccharides (LPS) in the serum improved. Also, an increasing abundance of *Ruminococcaceae* and a decreasing abundance of *Bacteroidaceae* was noticed. Interestingly, concentrations of acetic acid, propionic acid, butyric acid, valeric acid, and total SCFAs caused by CTX decreased. SCFAs increased with MP treatment in a dose-dependent manner, providing evidence of the prebiotic ability of MPs to prevent liver disease [93,100]. Hong et al. confirmed that MPs treatment could promote gut microbiota and reduce liver injury caused by CTX [100]. The results identified 18 and 29 endogenous metabolites in hepatic organs and feces, respectively, after mice were treated with MPs. At least eight metabolic pathways were involved with gut health and liver improvement, including taurine and hypotaurine metabolism, phenylalanine metabolism, *α*-linolenic acid metabolism, the tricarboxylic acid (TCA) cycle, phenylalanine, tyrosine, and tryptophan biosynthesis, arachidonic acid metabolism, and sphingolipid metabolism, as determined by ultra-high performance liquid chromatography with quadrupole time-of-flight mass spectrometry (UPLC-Q-TOF/MS). 

## 5. Application of Mesona Polysaccharides as Biomaterials for Medicine

Plant polysaccharides are nontoxic, highly stable, hydrophilic, biodegradable, and biocompatible, and they pose no significant negative side effects [70,71,101]. The beneficial improvements of MCPs and other polyphenolics were investigated for their bioavailability in drug carrier systems. Quercetin was loaded into smooth spherical nanoparticles, made from zein and Z–M NPs (zein–Mesona nanoparticles) under hydrophobic, hydrogen-bonding, and electrostatic interaction. The Z–M NPs showed a higher activity with regard to in vitro anti-inflammatory activity, NO (nitric oxide), TFN-*α*, IL-1*β*, and IL-6 in RAW 264.7 cells, compared with free quercetin [101]. Curcumin was loaded into zein–MCP nanoparticles (ZMC NPs) and measured under a simulated gastrointestinal environment. The results showed higher antioxidant activity and enhanced antitumor activity, by inducing cell apoptosis, against hepatocellular carcinoma cells (HepG-2), compared with free curcumin not encapsulated by nanoparticles [102]. Interestingly, MCPs and chitosan maintained the encapsulation curcumin effect. Curcumin was released at 7% after 2 h under simulated gastrointestinal tract conditions [103]. The results showed that using MPs to encapsulate the nanoparticles improved the retention of the bioactive compounds. Therefore, the Mesona plant showed promise as an alternative material for utilization in other fields, as well as in the food industry.

## 6. Conclusions and Future Prospects

*Mesona chinensis* possesses a proximate composition, physicochemical properties, nutraceuticals, phytochemistry, and pharmacological properties. Bioactive compounds contained in this plant have been widely used in Southern China and Southeast Asia as folk medicine and indigenous food. The Mesona plant has recently attracted interest due to its phytochemical and biological activities. High variations in chemical composition and functional properties are still challenging issues, which result from different varieties, origins, climate, harvesting times, storage, extraction processes, and determination methods. In the near future, the plant industry or smart farming will need to better understand and control this plant’s growth, harvesting time, and growing conditions. Research on the Mesona plant is mainly focused on *M. chinensis* from China. The geography of each growing area may alter plant characteristics and compositions; therefore, the uniqueness of Mesona varieties from different regions must be studied in detail, to enable its ultimate utilization in functional ingredients, pharmaceuticals, and other fields, according to matched proximate compositions. The link between polysaccharide extracts and gut health is interesting and has implications for corresponding physical and mental disorders. However, meta-analysis data and safety concerns must be further investigated before therapeutic applications and innovation processes are introduced.

## Figures and Tables

**Figure 1 foods-13-01134-f001:**
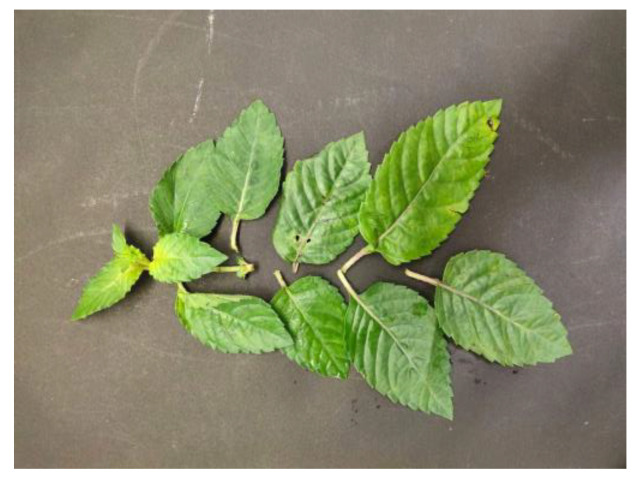
*Mesona chinensis* leaves.

**Figure 2 foods-13-01134-f002:**
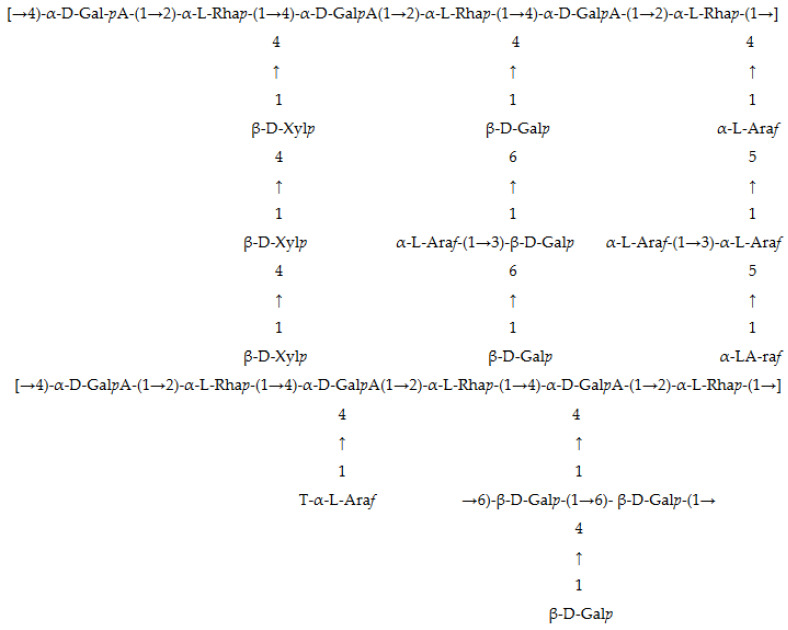
A possible acidic polysaccharide structural model of *M. chinensis*.

**Figure 3 foods-13-01134-f003:**
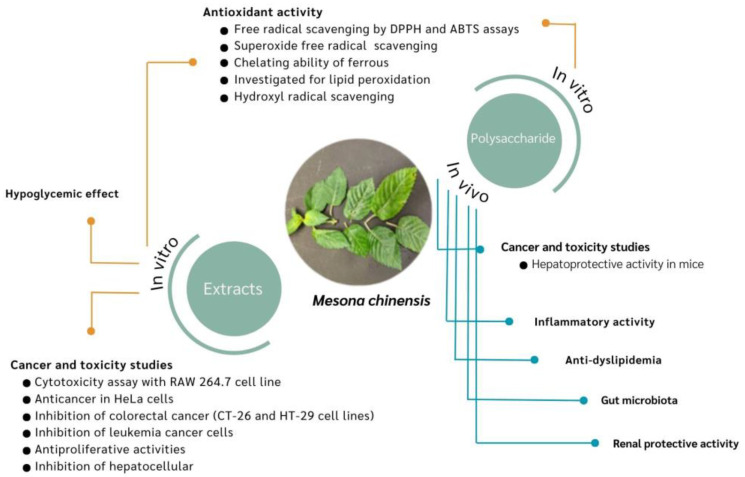
Summary of proposed pharmacological properties of Mesona polysaccharides (MPs).

**Table 1 foods-13-01134-t001:** Proximate composition of *Mesona chinensis* leaves from different regions of South China and Taiwan (China).

Source	Extraction	Proximate Composition (%)				Refs.
		Crude Protein	Crude Fat	Crude Fiber	Ash	
*Mesona chinensis* leaf powder (China)	With sodium bicarbonate in heated water at 95 °C for 2 h	9.74	-	2.98	30.9	[36]
*Mesona chinensis* powder (China)	Water extract	9.0	0.1	-	28.2	[38]
*Mesona chinensis* leaves (farm market in Taiwan)	With sodium bicarbonate in heated water at 95 °C for 4 h	4.56	-	1.07	26.97	[39]
*Mesona chinensis* leaves (contracted farmer, Taiwan)	With sodium bicarbonate in heated water at 95 °C for 4 h	10.04	0.52	1.47	26.2	[40]
*Mesona chinensis* leaves (contracted farmer in Miao-Li, Taiwan)	With sodium bicarbonate in heated water at 95 °C for 4 h	4.60	0.90	1.10	27.0	[41]

**Table 2 foods-13-01134-t002:** Physicochemical properties of *Mesona chinensis* polysaccharides.

Collected Region	Extraction	Yield (%)	Molecular Weight (kDa)	Chemical Composition (%)			Monosaccharide Composition (Mole Ratios)							Refs.
				Total Sugar	Uronic Acid	Protein	Glu	Gal	Gala	Rha	Ara	Man	Xyl	
Xiaoshicheng, Ganzhou, Jiangxi, China	Boil in hot water 95 °C for 2 h with Na_2_CO_3_	-	158	29.03	17.06	22.64	N.D.	2.80	2.40	N.D.	N.D.	N.D.	5.50	[4]
China	Boil in hot water 95 °C for 2 h with Na_2_HCO_3_	29.36	16.26	42.20	13.80	9.74	2.30	3.10	1.40	1.20	2.30	0.20	1.00	[36]
Xiaoshicheng, Jiangxi, China	Boil in hot water 95 °C for 3 h with Na_2_CO_3_	-	141.6	16.88	36.91	-	1.36	3.76	17.5	0.87	0.14	-	2.00	[48]
Ganzhou, Jiangxi, China.	Boil in hot water 95 °C for 2.5 h	-	375	81.12	-	14.07	4.90	2.16	6.75	1.38	1.64	-	0.42	[49]
Xiaoshicheng, Jiangxi, China	Boil in hot water 90 °C for 2 h with Na_2_CO_3_	7.05	1450	-	29.30	10.40	1.38	1.0	-	-	-	-	-	[50]
Yichun, Jiangxi, China	Boil in hot water 100 °C	0.84	44.39	30.69	20.86	25.30	1.12	1.97	1.69	0.42	0.30	0.50	N.D.	[51]
Xiaoshicheng, Ganzhou, Jiangxi, China	Boil in hot water 95 °C for 2.5 h with Na_2_CO_3_	11.14	195	34.40	24.30	17.30	1.00	1.34	0.25	-	N.D.	-	N.D.	[52]
Xiaoshicheng, Ganzhou, Jiangxi, China	Boil in hot water 95 °C for 2.5 h with Na_2_CO_3_	-	204	32.28	29.52	31.35	6.24	0.82	-	0.11	0.32	-	0.34	[54]
Ganzhou, Jiangxi, China.	Boil in hot water 100 °C for 2 h	1.68	157	39.01	29.30	27.52	1.49	0.68	6.33	-	-	-	2.54	[55]

N.D.: Not detectable or lower than the limit of determination. Glu: glucose, Gal: galactose, Gala: galacturonic acid, Rha: rhamnose, Ara: arabinose, Man: mannose, and Xyl: xylose.

## Data Availability

No new data were created or analyzed in this study. Data sharing is not applicable to this article.

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
