# Peer review of "Functionality and Health-Promoting Properties of Polysaccharide and Plant-Derived Substances from Mesona chinensis"

_foods, 2024, doi:10.3390/foods13071134_

Round 1

Reviewer 1 Report

Comments and Suggestions for Authors

The topic is interesting and provides some insights into the polysaccharide composition and pharmacology of Mesona chinensis, however it absolutely needs to be reviewed as it is poorly written. Additionally, although consumed as a functional drink and jelly type dessert, in my opinion, the scope seems more appropriate for other journals including "Plants" rather than food journal.

Further comments are provided below.

To prevent confusion, use the same synonym for the plant consistently throughout the text.

It could be beneficial to integrate other informations more seamlessly into the discussions. Indeed, there are at least twenty recent references (2022-2024) that are not included in the document.

Additionally, a paragraph on toxicity studies of the plant or its metabolites should be added.

The extraction and phytochemistry section should be separated from the pharmacology section.

A figure displaying a photograph of the plant is required.

A figure summarizing the pharmacological properties is required,

Figure 1 does not add any benefit to the document.

Correct the incomplete or incorrect references, for example, references 43, 87..., add doi for all references, as the majority of references are missing doi.

Many paragraphs are complexes and should be rephrased, or contain typos or grammatical errors

Here are some few examples :

Lines 75-77 correct

Line 36 correct taiwan instead tiwan

correct line 40

line 43 : rephrase the paragraph

rephrase lines 67-68

Comments on the Quality of English Language

Manuscript is poor in English writing

Author Response

Comments 1: To prevent confusion, use the same synonym for the plant consistently throughout the text.

Response 1: Thank you for pointing this out. I agree with this comment. Therefore, I have been adjusting the plant name in the same synonym consistently throughout the text as below.

-          Page on 3rd in table 1 at line 85.

-          Page on 6th in the 1st paragraph at line 150, 153, 163 and 170.

-          Page on 7th in the 1st paragraph at line 214.

-          Page on 8th in the 3rd paragraph at line 270.

-          Page on 8th in the 4th paragraph at line 290.

-          Page on 9th in the 2nd paragraph at line 307, 310 and 311.

-          Page on 10th in the 3rd paragraph at line 359.

-          Page on 10th in the 4th paragraph at line 375 and 381.

Comments 2: It could be beneficial to integrate other informations more seamlessly into the discussions. Indeed, there are at least twenty recent references (2022-2024) that are not included in the document.

Response 2: I have added more information that used involved reference during 2022-2024 as shown on page 11th in the 2nd paragraph at line 426-429 and 3rd paragraph at line 433.

Comments 3:  Additionally, a paragraph on toxicity studies of the plant or its metabolites should be added.

Response 3: I have already done on page 8th in the 3rd paragraph at line 264.

Comments 4: The extraction and phytochemistry section should be separated from the pharmacology section.

Response 4: I have already done on page 6th in the 2nd paragraph at line 178.

Comments 5: A figure displaying a photograph of the plant is required.

Response 5: I have already done on page 2nd at line 71.

Comments 6: A figure summarizing the pharmacological properties is required

Response 6: I have already done on page 7th at figure 3.

Comments 7: Figure 1 does not add any benefit to the document.

Response 7: I have already deleted and replace with a photograph of the plant.

Comments 8: Correct the incomplete or incorrect references, for example, references 43, 87..., add doi for all references, as the majority of references are missing doi.

Response 8: I have already added DOI for all references.

Comments 9: Many paragraphs are complexes and should be rephrased, or contain typos or grammatical errors

Response 9: I have already done as below

-          Page on 1st in the abstract.

-          Page on 1st in the 2nd paragraph at line 43 and 45.

-          Page on 2nd in the 1st paragraph at line 46 and 52-54.

-          Page on 2nd in the 1st paragraph at line 57-65.

-          Page on 3rd in the 3rd paragraph at line 104-106.

-          Page on 4th in the 1st paragraph at line 118-119.

-          Page on 6th in the 1st paragraph at line 151, 162 and 168.

-          Page on 6th in the 2nd paragraph at line 184, 187.

-          Page on 7th in the 2nd paragraph at line 213-214, 216-217.

-          Page on 8th in the 2nd paragraph at line 256.

-          Page on 8th in the 3rd paragraph at line 265-266 and 279-286.

-          Page on 9th in the 2nd paragraph at line 310 and 312-313.

-          Page on 10th in the 4th paragraph at line 370 and 372.

-          Page on 11th in the 2nd paragraph at line 423 and 426-429.

-          Page on 12th in the 2nd paragraph at line 468 and 469.

Reviewer 2 Report

Comments and Suggestions for Authors

In this manuscript “Functionality and Pharmacological Activities of Polysaccharide and Bioactive Compounds from Mesona chinensis”, the review revealed proximate composition, polysaccharide type and pharmacological property of M. chinensis extract. The future prospect is required to study mechanism of the plant extract in vitro/in vivo, and gut health effect.

The review demonstrates clarity and comprehensiveness, addressing relevant gaps in knowledge within the field. It provides valuable insights that contribute to the scientific discourse. Despite the presence of a recent similar review, this current one remains pertinent and intriguing to the scientific community, offering fresh perspectives and updated information. The cited references predominantly stem from recent publications, ensuring relevance and currency within the discourse. Furthermore, the review maintains coherence in its statements and draws well-supported conclusions from the listed citations. The figures, tables, and other visual aids effectively illustrate the data, enhancing understanding and interpretation for the readers.

The research is meaningful, but there are some problems with your manuscript. The comments and problems are as follows:

1.I think the abstract should be revised to make the article more logical and to highlight the research purposes and significance of the review.

2.At the end of the cited part of the manuscript, the content can be appropriately added to highlight the research focus and innovation.

3.The introduction is out of order. I suggest that the topic of each paragraph should be clear, the relevant chapters can be put together, and the context or evidence should be described logically.

4.The discussion section is too small. It is recommended to rewrite the author's discussion section.

Author Response

Comments 1: think the abstract should be revised to make the article more logical and to highlight the research purposes and significance of the review.

Response 1: Thank you for pointing this out. I agree with this comment. Therefore, I have already done as shown in the abstract.

Comments 2: At the end of the cited part of the manuscript, the content can be appropriately added to highlight the research focus and innovation.

Response 2: I have added more information as shown on page 11th in the 3rd paragraph at line 433.

Comments 3:  The introduction is out of order. I suggest that the topic of each paragraph should be clear, the relevant chapters can be put together, and the context or evidence should be described logically.

Response 3: The introduction has been revised.

Comments 4: The discussion section is too small. It is recommended to rewrite the author's discussion section.

Response 4: I have already done as below

-          Page on page 3rd in the 3rd paragraph at line 107 and 115.

-          Page on page 4th in the 1st paragraph at line 117-121.

-          Page on page 6th in the 1st paragraph at line 175-177.

-          Page on page 6th in the 2nd paragraph at line 189-190.

-          Page on page 8th in the 1st paragraph at line 246-248.

-          Page on page 8th in the 2nd paragraph at line 261-263.

-          Page on page 9th in the 1st paragraph at line 296-299.

-          Page on page 9th in the 3rd paragraph at line 328-330.

-          Page on page 10th in the 2nd paragraph at line 355-357.

-          Page on page 10th in the 4th paragraph at line 386-388.

Reviewer 3 Report

Comments and Suggestions for Authors

1.     Mesona chinensis (Chao Kuay) is belonging to the Lamiaceae family. It is mainly distributed  in southern China, Taiwan and Southeast Asia such as Thailand, Malaysia, Philippines, Indonesia, and Vietnam. The author lists Taiwan as a separate country, which is unacceptable to Chinese reviewers. Please strictly abide by the basic common sense of one China. Although scientific research does not involve political factors, the author's basic common sense needs to be standardized.

2.     There are multiple grammatical errors throughout the text that need to be carefully reviewed and corrected.

3.     Many sentences are overly long, consider breaking them into shorter sentences to improve readability.

4.     The introduction needs to provide a stronger background and clearer research objectives.

5.     The descriptions of morphological and chemical characteristics of Mesona chinensis are too brief.

6.     The discussion and analysis of key data are not thorough and comprehensive enough.

7.     Some professional terms lack definitions or explanations.

8.     Transitions between sections are not natural and fluent, proper connecting phrases are needed.

Comments on the Quality of English Language

1.     Mesona chinensis (Chao Kuay) is belonging to the Lamiaceae family. It is mainly distributed  in southern China, Taiwan and Southeast Asia such as Thailand, Malaysia, Philippines, Indonesia, and Vietnam. The author lists Taiwan as a separate country, which is unacceptable to Chinese reviewers. Please strictly abide by the basic common sense of one China. Although scientific research does not involve political factors, the author's basic common sense needs to be standardized.

2.     There are multiple grammatical errors throughout the text that need to be carefully reviewed and corrected.

3.     Many sentences are overly long, consider breaking them into shorter sentences to improve readability.

4.     The introduction needs to provide a stronger background and clearer research objectives.

5.     The descriptions of morphological and chemical characteristics of Mesona chinensis are too brief.

6.     The discussion and analysis of key data are not thorough and comprehensive enough.

7.     Some professional terms lack definitions or explanations.

8.     Transitions between sections are not natural and fluent, proper connecting phrases are needed.

Author Response

Comments 1: The author lists Taiwan as a separate country, which is unacceptable to Chinese reviewers. Please strictly abide by the basic common sense of one China. Although scientific research does not involve political factors, the author's basic common sense needs to be standardized.

Response 1: Thank you for pointing this out. I agree with this comment. Therefore, I have already done as below

-          Page on 1st in the abstract at line 11.

-          Page on 1st in the introduction at line 38 and 39.

-          Page on 3rd in the 2nd paragraph at line 112.

Comments 2: There are multiple grammatical errors throughout the text that need to be carefully reviewed and corrected.

Response 2: I have already done as below

-          Page on 1st in the abstract.

-          Page on 1st in the 2nd paragraph at line 43 and 45.

-          Page on 2nd in the 1st paragraph at line 46 and 52-54.

-          Page on 2nd in the 1st paragraph at line 57-65.

-          Page on 3rd in the 3rd paragraph at line 104-106.

-          Page on 4th in the 1st paragraph at line 118-119.

-          Page on 6th in the 1st paragraph at line 151, 162 and 168.

-          Page on 6th in the 2nd paragraph at line 184, 187.

-          Page on 7th in the 2nd paragraph at line 213-214, 216-217.

-          Page on 8th in the 2nd paragraph at line 256.

-          Page on 8th in the 3rd paragraph at line 265-266 and 279-286.

-          Page on 9th in the 2nd paragraph at line 310 and 312-313.

-          Page on 10th in the 4th paragraph at line 370 and 373.

-          Page on 11th in the 2nd paragraph at line 423 and 426-429.

-          Page on 12th in the 2nd paragraph at line 468 and 469.

Comments 3: Many sentences are overly long, consider breaking them into shorter sentences to improve readability.

Response 3: I have already done as below

-          Page on 2nd in the 1st paragraph at line 52-53.

-          Page on 3rd in the 3rd paragraph at line 104-106.

-          Page on 6th in the 1st paragraph at line 151 and 170

-          Page on 8th in the 2nd paragraph at line 256.

-          Page on 8th in the 3rd paragraph at line 279-286.

-          Page on 9th in the 2nd paragraph at line 310 and 312-313.

Comments 4: The introduction needs to provide a stronger background and clearer research objectives.

Response 4: I have already revised in the abstract and introduction.

Comments 5: The descriptions of morphological and chemical characteristics of Mesona chinensis are too brief

Response 5: I have already done as shown on page 1st in the 2nd paragraph at line 41-43.

Comments 6: The discussion and analysis of key data are not thorough and comprehensive enough.

Response 6: I have already done as below

  • Page on page 3rd in the 3rd paragraph at line 107 and 115.
  • Page on page 4th in the 1st paragraph at line 117-121.
  • Page on page 6th in the 1st paragraph at line 175-177.
  • Page on page 6th in the 2nd paragraph at line 189-190.
  • Page on page 8th in the 1st paragraph at line 246-248.
  • Page on page 8th in the 2nd paragraph at line 261-263.
  • Page on page 9th in the 1st paragraph at line 296-299.
  • Page on page 9th in the 3rd paragraph at line 328-330.
  • Page on page 10th in the 2nd paragraph at line 355-357.
  • Page on page 10th in the 4th paragraph at line 386-388.

Comments 7: Some professional terms lack definitions or explanations.

Response 7: I have already done as below

  • Page on page 3rd in the 3rd paragraph at line
  • Page on page 5th in table 2 at line 148.
  • Page on page 10th in the 5th paragraph at line 390.

Comments 8: Transitions between sections are not natural and fluent, proper connecting phrases are needed.

Response 8: I have already done as below

  • Page on 1st in the 2nd paragraph at line 43.
  • Page on 2nd in the 2nd paragraph at line 55.
  • Page on page 3rd in the 3rd paragraph at line 113-115.
  • Page on page 6th in the 2nd paragraph at line
  • Page on page 9th in the 3rd paragraph at line

Reviewer 4 Report

Comments and Suggestions for Authors

Chinese mesona, Mesona chinensis, grows extensively in East Asia such as south east China, Japan and Taiwan preferring ravines, grassy, dry, and sandy areas. The plants contain phenols, tannins, and flavonoids. In regard to biological activity, the plant extracts have been reported to induce anti-diabetic, anti-cancer, and anti-diarrhea effects in pre-clinical research, all of which are possible due to the strong antioxidant nature of the extracts.

In connection with supplementing the research on the plant species Mesona chinensis, I appreciate the clear information on natural substances of polysaccharide type. In an original scientific review, 59 phytochemical compounds and their biological activities were demonstrated.

The introduction of the article is very general focused on global topics around plant species.

Of course, it is sufficient raw material for various types of industry (pharmaceutical, food, beverage …). The data are supplemented with proximate and mineral composition, polysaccharide of Mesona chinensis extraction and their pharmacological properties with antioxidant activity, cancer and toxicity studies, hypolipidemic and hypoglycemic effects, renal protective and inflammatory activities.

The literature citations are adequate enough and finally the manuscript does not contain any irrelevant information’s. The very important is comparison of the information, knowledges and results from foreign literature and various literature sources.

Conclusion is very general. I agree with the authors that there is a need for constant monitoring of new knowledges (relationship between therapeutic application and innovation).

In regard to my opinion the contents of the manuscript in line with policy of the journal, the text has been prepared according to the format and style of journal including the body of the manuscript, page size and referencing.

Comments on the Quality of English Language

Minor editing of English language requiredMinor editing of English language required

Author Response

Comments 1: Minor editing of English language required Minor editing of English language required.

Response 1: Thank you for pointing this out. I agree with this comment. Therefore, I have already done as below

-          Page on 1st in the 2nd paragraph at line 43 and 45.

-          Page on 2nd in the 1st paragraph at line 46.

-          Page on 6th in the 1st paragraph at line 151, 162, 168 and 170.

-          Page on 6th in the 2nd paragraph at line 184, 187.

-          Page on 7th in the 2nd paragraph at line 213-214, 216-217.

-          Page on 8th in the 4th paragraph at line 265 and 266.

-          Page on 9th in the 2nd paragraph at line 310 and 312-313.

-          Page on 10th in the 4th paragraph at line 370 and 373.

-          Page on 11th in the 2nd paragraph at line 423

-          Page on 12th in the 2nd paragraph at line 468 and 469.

Round 2

Reviewer 1 Report

Comments and Suggestions for Authors

- correct line 198: p-hydroxybenzoic acid,

- line 221: p-hydroxybenzoic acid,

- arrange the table 2

- in conclusion: rephrase lines 539-542

Reviewer 2 Report

Comments and Suggestions for Authors

Accepted.

Reviewer 3 Report

Comments and Suggestions for Authors

1.       It is strongly recommended that the author remove the revision mode and only keep the final version, with modifications highlighted in red.

2.       At present, the repetition rate is relatively high. It is recommended that the author reduce the repetition rate, otherwise the manuscript may be considered rejected.

Comments on the Quality of English Language

None
